# Optimization of machine tool processing scheduling based on differential evolution algorithm

**Yuehong Zhang[1], Mianhao Zhang [2]***

**1** Jinhua University of Vocational Technology, Jinhua, Zhejiang, China, **2** Zhejiang Normal University, Jinhua, Zhejiang, China

\* zhangmh_edu@163.com

## Abstract

Machine tool processing scheduling plays a pivotal role in modern manufacturing systems, significantly influencing production efficiency, resource utilization, and timely delivery. Due to its combinatorial and NP-hard characteristics, traditional optimization techniques often face challenges when dealing with large-scale and complex scheduling problems. In this paper, we present an optimization approach for machine tool scheduling that leverages the Differential Evolution (DE) algorithm. By tailoring DE for discrete scheduling environments through specialized encoding and decoding techniques, the algorithm is able to effectively explore the solution space while ensuring the generation of feasible schedules. The results from our experiments reveal that the proposed approach outperforms conventional heuristic methods, particularly in minimizing makespan and achieving a balanced workload distribution across machines. This study underscores the potential of DE as a robust, adaptive, and efficient optimization tool for tackling complex scheduling problems in the context of intelligent manufacturing systems.

## 1 Introduction

With the rapid progress of intelligent manufacturing and the ongoing evolution of Industry 4.0 technologies [1–3], the manufacturing sector is undergoing a significant transformation—from conventional mass production systems to more agile, flexible, and customer-oriented production models. This paradigm shift has introduced new opportunities for efficiency and innovation, but it has also heightened the complexity of managing production processes. In this context, the optimization of machine tool scheduling has emerged as a cornerstone of modern manufacturing operations, playing a vital role in enhancing productivity, optimizing resource utilization, and ensuring timely order fulfillment [4–7]. Nevertheless, the increasing intricacy and dynamic nature of manufacturing environments have made scheduling problems more challenging than ever, necessitating advanced optimization techniques and intelligent decision-making frameworks.

As manufacturing systems continue to expand in scale and complexity, scheduling problems have become increasingly high-dimensional and computationally intensive. In large-scale, customized production environments, scheduling systems must be capable of

**Data availability statement:** All relevant data are within the manuscript and its Supporting information files.

**Funding:** The author(s) received no specific funding for this work.

efficiently managing hundreds to thousands of job operations within tight production timelines. This significant increase in scheduling scope not only intensifies computational demands but also complicates the search for optimal or near-optimal solutions. Furthermore, modern manufacturing processes are characterized by dynamic and unpredictable elements, including sudden equipment breakdowns, supply chain disruptions, last-minute order modifications, and emergency job insertions. These uncertainties are no longer occasional but rather persistent features of real-world production, requiring scheduling algorithms to be highly adaptive, responsive, and capable of real-time decision-making. In addition, the objectives of manufacturing scheduling have evolved from single-goal optimization to balancing a diverse and often conflicting set of goals. Production systems are now expected to simultaneously minimize delivery times, reduce energy consumption, maximize resource utilization, and enhance overall sustainability [8–10]. The presence of these competing objectives presents significant trade-offs, making it increasingly difficult for traditional methods to generate balanced and effective solutions. Conventional scheduling techniques, such as mathematical programming approaches [11–15] and heuristic-based strategies [16–18], have demonstrated clear limitations in addressing these multifaceted challenges. Their drawbacks include poor scalability due to the NP-hard nature of scheduling problems, limited responsiveness to dynamic conditions stemming from their reliance on static assumptions, and inadequate multi-objective optimization performance, often failing to identify Pareto-optimal trade-offs across conflicting goals. As a result, there is a growing consensus in both academia and industry that the future of scheduling lies in the development of intelligent optimization algorithms that are not only computationally efficient but also robust, scalable, and capable of navigating complex, dynamic, and multi-objective problem spaces [19–21]. These algorithms offer the potential to fundamentally enhance decision-making processes in smart manufacturing environments, aligning with the broader goals of Industry 4.0 and beyond.

In recent years, as intelligent manufacturing and digital transformation continue to advance, manufacturing systems have encountered increasingly complex scheduling problems [22–24]. These challenges arise from several factors, including larger production scales, a growing number of constraints, and more diverse optimization objectives. In response to these difficulties, swarm intelligence optimization algorithms have gained significant attention in the field of manufacturing scheduling. These algorithms are particularly valued for their excellent global search capabilities, adaptive search strategies, and strong robustness in uncertain and dynamic environments [25–27]. Unlike traditional optimization methods that often rely on precise mathematical models and detailed analytical derivations, swarm intelligence algorithms do not require a rigid mathematical formulation of the problem. Instead, they draw inspiration from natural group behaviors—such as biological evolution, social cooperation, and swarm migration—to perform complex searches. This approach enables them to offer greater flexibility and adaptability, making them particularly effective in solving high-dimensional, discrete, dynamic, and multi-constrained scheduling problems.

Among various swarm intelligence methods [28], the Differential Evolution (DE) algorithm stands out for its simple structure, few control parameters, ease of implementation, and high search efficiency [29–35]. It has shown excellent performance in numerous engineering optimization domains, such as nonlinear function optimization, structural parameter design, and image recognition. The core mechanism of DE consists of three main operations: mutation, which enhances population diversity and exploration through differential strategies; crossover, which introduces variation while preserving good genetic information; and selection, which guides the population's evolution based on fitness values. This mechanism not only improves the algorithm's adaptability to complex search spaces but also effectively mitigates premature convergence, making DE especially suitable for solving non-convex,

multimodal, and highly constrained problems. Compared with other classical evolutionary strategies such as Genetic Algorithms (GA) [36–41] and Particle Swarm Optimization (PSO) [42–47], DE eliminates the need for complex encoding and decoding processes, shows low dependence on the quality of the initial population, and demonstrates better convergence stability and search efficiency in high-dimensional solution spaces. Therefore, DE is widely regarded as a competitive global optimization tool with strong theoretical value and broad industrial application potential. However, despite the theoretical breakthroughs and preliminary successes of DE in various scheduling scenarios, challenges remain when applying it to real-world manufacturing environments. These include dynamic disturbances (e.g., urgent job insertions, resource changes), multi-resource coupling, constraint coordination, and multi-objective trade-offs. In such contexts, the standard DE algorithm still suffers from limited adaptability and optimization precision. Thus, there is an urgent need to structurally enhance and tailor the DE algorithm to address the specific characteristics of complex scheduling problems. Doing so will enable higher-level optimization performance and provide stable, efficient, and scalable decision-making support for intelligent manufacturing systems.

Based on the above literature, this paper aims to address the limitations of traditional optimization methods in solving complex machine tool processing scheduling problems by proposing a novel approach grounded in the DE algorithm. While previous studies have demonstrated the potential of evolutionary algorithms in optimization tasks, few have effectively adapted DE for discrete, multi-objective scheduling in real-world manufacturing scenarios. The key innovations of this study are as follows:

1. Unlike standard DE designed for continuous domains, this paper introduces a problem-specific encoding and decoding mechanism that allows DE to operate effectively in discrete scheduling environments. This ensures that each solution generated corresponds to a valid and feasible job-machine assignment.
2. By leveraging the global search capability of DE and tailoring its operators to the scheduling context, the algorithm demonstrates strong performance even in the presence of combinatorial complexity and resource constraints.
3. The study conducts extensive simulations and comparative experiments against conventional heuristic methods, illustrating that the proposed DE-based approach yields more competitive and reliable scheduling results.
4. The findings offer both theoretical insights and practical guidance for the integration of intelligent metaheuristic optimization techniques in advanced manufacturing systems. They present a scalable and efficient solution that can be applied to future industrial scheduling challenges, enhancing the flexibility and performance of manufacturing operations.

This paper is structured as follows: Sect 1 introduces the research background, significance, and reviews related studies both domestically and internationally. Sect 2 presents the fundamental principles and standard procedure of the DE algorithm, along with an analysis of how parameter settings affect its performance. Sect 3 verifies the effectiveness of the DE algorithm in global optimization through an experiment on the Rosenbrock function. Sect 4 applies the DE algorithm to a practical machine tool scheduling problem by constructing a profit-maximization model and conducting simulation verification. Finally, Sect 5 summarizes the main findings and proposes future research directions, including multi-machine scheduling, optimization under dynamic uncertainty, and multi-objective extensions.

## 2 Machine tool modeling and problem description

In modern manufacturing environments, efficient and adaptive scheduling of machine tool operations is crucial for meeting delivery deadlines, reducing operational costs, and responding to dynamic production changes. However, many existing models either focus solely on theoretical scheduling constructs or assume idealized machine conditions that do not capture real-world constraints such as task-switching overhead, failure uncertainty, and profit-based objectives. The proposed model addresses these gaps by integrating practical features—such as constant switching times and probabilistic breakdown modeling—while retaining computational tractability. This balance between realism and simplicity ensures the model is both applicable in real production systems and compatible with intelligent optimization algorithms. Therefore, constructing this model is essential to bridge the gap between abstract scheduling theory and actual manufacturing practice.

### 2.1 Simplification and assumption of modeling

**Assumption 1:** Although machine tools processing may generate micro-level vibrations during operation, which can affect surface quality, this study focuses on task-level time scheduling. Therefore, the influence of such micro-vibrations is considered negligible and excluded from the scheduling model, simplifying the estimation of processing time and energy consumption.

**Assumption 2:** Task switching on a machine typically involves preparatory operations such as tool changes or repositioning. For modeling simplicity, we assume a constant switching time between tasks, regardless of task types. This allows the construction of a standardized task processing cycle and reduces scheduling complexity.

**Assumption 3:** To capture random failures in machine operations, a Poisson distribution is used to model the frequency of machine breakdowns. This statistical assumption simplifies the dynamic rescheduling logic and provides a realistic yet tractable representation of machine availability fluctuations.

These assumptions help control the complexity of the scheduling model without compromising key features, thereby facilitating the design and optimization of the proposed algorithm.

### 2.2 Model necessity analysis

A machine tool is a powered mechanical device used to shape or machine metal or other rigid materials by cutting, drilling, grinding, or otherwise removing material. These machines are essential in manufacturing processes and are commonly used to produce parts with high precision and repeatability, as shown in Fig 1 . The image shows CNC machine tools, which are controlled by computer programs, allowing for automated, accurate, and efficient machining operations. Different types of machine tools include lathes, milling machines, and machining centers, each designed for specific tasks within the production line.

Machine tool scheduling optimization is not merely an abstract mathematical problem; at its core, it aims to address the practical challenge of efficiently utilizing machine resources in real manufacturing environments. Through systematic modeling, the complex and dynamic elements of production can be structured and quantified, providing a solid foundation for effective scheduling and optimization strategies. Specifically, a well-constructed scheduling model must incorporate the following key dimensions: - Diversity of machine tool types:

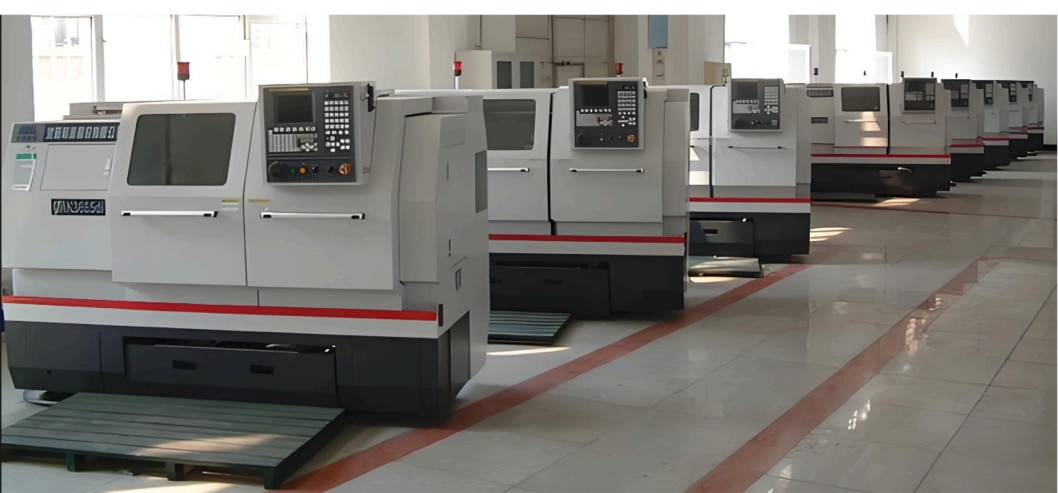

**Fig 1. Schematic diagram of machine tool.**

Different types of machining equipment-such as CNC lathes, vertical machining centers, and multiaxis machines-exhibit significant differences in processing capabilities, applicable processes, and operational responsiveness. These differences directly impact task-machine matching strategies and scheduling priorities. - Dynamic processing characteristics:

Critical parameters such as machining time, energy consumption, equipment wear, and failure rates are inherently dynamic and uncertain. Accurate representation of these factors in the model enhances the realism and robustness of the resulting scheduling solutions. - Complex constraint conditions:

Real-world scheduling is subject to numerous constraints, including machine capacity limitations, job priority levels, and heterogeneity or compatibility among machines. These constraints define the feasibility and quality of scheduling solutions and must be rigorously incorporated into the optimization process.

In summary, constructing a scheduling model that faithfully captures the operational logic of the manufacturing floor while remaining amenable to algorithmic optimization is essential for achieving efficient and intelligent production scheduling.

In the study of machine tool scheduling based on the Differential Evolution (DE) algorithm, we consider a flexible manufacturing cell (FMC) comprising $N$ heterogeneous machine tools, each with distinct operational characteristics. To accurately reflect real-world manufacturing scenarios, the scheduling problem is formulated under the following modeling assumptions:

**Static properties of machines:** Each machine $M_k$ is characterized by its specific static attributes, including a unique processing speed $v_k$, maximum load capacity $W_{\max,k}$, and energy consumption coefficients—namely, idle power $P_{\mathrm{idle},k}$ and cutting power $P_{\mathrm{cut},k}$. These parameters capture the heterogeneity of the machines in terms of productivity, load-bearing capacity, and energy efficiency.

**Dynamic task processing time:** The processing time $T_{ik}$ required for task $J_i$ on machine $M_k$ is influenced by both the task's properties and the machine's performance. It is defined by the following expression [48]:

$$T_{ik} = \alpha \cdot \frac{L_i}{v_k} + \beta \cdot H_i \tag{1}$$

where $L_i$ represents the processing length of task $J_i$, $H_i$ denotes the material hardness, and $\alpha$, $\beta$ are empirical coefficients used to reflect the relative influence of machining speed and material characteristics.

**Multi-objective optimization:** The scheduling objective includes simultaneous minimization of the makespan and the total energy consumption, formulated as:

$$E_{\text{total}} = \sum_{k=1}^{N} \left( P_{\text{idle},k} \cdot t_{\text{idle},k} + P_{\text{cut},k} \cdot t_{\text{cut},k} \right) \tag{2}$$

This multi-objective design reflects the industry's need for both high productivity and energy-efficient operation, aligning with the goals of sustainable and green manufacturing.

**Precedence constraints among tasks:** Some tasks must follow a predefined execution order due to process dependencies (e.g., in assembly operations), such that task $J_j$ cannot start until task $J_i$ is completed.

**Machine load constraints:** At any given time $t$, the total load on machine $M_k$, denoted $W_k(t)$, must not exceed its maximum allowable load $W_{\text{max},k}$, ensuring safe and stable operation.

## 2.3 Objective of this paper

The primary objective of this study is to develop an efficient and robust scheduling optimization method for machine tool operations in flexible manufacturing environments, based on the Differential Evolution (DE) algorithm. By leveraging the global search capability and adaptability of DE, the study aims to address the challenges posed by heterogeneous machine resources, dynamic processing conditions, and multi-objective trade-offs. Specifically, the research seeks to:

1. Construct a scheduling model that accurately reflects real-world machine characteristics, including varying processing speeds, energy consumption, and operational constraints.
2. Design a discrete-adapted DE algorithm with specialized encoding and decoding strategies suitable for complex job-machine assignment problems.
3. Incorporate multi-objective optimization to simultaneously minimize makespan and energy consumption, while satisfying precedence and load constraints.
4. Validate the proposed approach through simulation experiments to demonstrate its effectiveness, scalability, and superiority over conventional heuristics.

Through these goals, the study aims to provide a practical and intelligent solution for optimizing machine tool processing scheduling in modern manufacturing systems under the Industry 4.0 paradigm.

## 3 Optimization of machine tool processing scheduling based on DE algorithm

To effectively address the complexity and multi-constraint nature of machine tool processing scheduling, this section presents a scheduling optimization framework based on the Differential Evolution (DE) algorithm. The section is structured as follows: First, we introduce the fundamentals and advantages of the DE algorithm in the context of scheduling problems. Then, we describe the standard DE procedure in detail, including its mutation, crossover, and selection mechanisms. Finally, we outline the basic flow of the DE-based scheduling

optimization process, laying the groundwork for the algorithm's adaptation to discrete scheduling tasks in subsequent sections. Through this approach, we aim to demonstrate the algorithm's capability to handle high-dimensional, multi-objective scheduling problems typical in intelligent manufacturing environments.

## 3.1 The proposal of DE algorithm

The Differential Evolution (DE) algorithm was first introduced by Storn and Price in 1995 as a simple yet powerful population-based evolutionary algorithm designed for global optimization over continuous domains [49]. Its core mechanism revolves around three main operators: mutation, crossover, and selection, which are used iteratively to evolve a population of candidate solutions toward the global optimum. Unlike traditional evolutionary algorithms, DE utilizes the scaled difference of randomly selected individuals to perturb existing solutions, allowing it to balance exploration and exploitation effectively. Its advantages include a minimal number of control parameters, ease of implementation, and strong global convergence performance, which have led to its wide adoption across various scientific and engineering domains [50].

In recent years, DE has been increasingly applied to combinatorial and discrete optimization problems, including the complex and NP-hard domain of machine tool scheduling [51–53]. In the context of modern manufacturing systems, machine tool scheduling involves assigning a sequence of tasks to heterogeneous machines under various operational constraints, such as processing time, machine availability, energy consumption, and task precedence. Traditional methods—such as integer programming, dispatching rules, and classical heuristics—often struggle to cope with the scale and dynamic nature of real-world scheduling scenarios.

To bridge this gap, researchers have adapted the DE algorithm to suit discrete job scheduling environments by designing specialized encoding and decoding mechanisms, enabling it to represent job sequences and machine-task assignments effectively. Additionally, variants of DE have been developed to handle multi-objective scheduling problems, where objectives like minimizing makespan, reducing energy consumption, and balancing machine workloads must be optimized simultaneously.

The application of DE to machine tool processing scheduling has shown promising results in both simulation and practical implementations. Its ability to escape local optima and maintain solution diversity makes it particularly suitable for flexible manufacturing systems, where machine heterogeneity, dynamic disturbances, and complex constraint interactions are common. As a result, DE has become a valuable tool for addressing scheduling challenges in intelligent manufacturing and Industry 4.0-driven production environments.

## 3.2 Standard differential evolution algorithm

The DE algorithm is a global optimization method rooted in the theory of swarm intelligence. It simulates the cooperative and competitive interactions among individuals within a population to guide the search process toward the global optimum. DE employs a real-valued encoding scheme and features a simple yet effective evolutionary process composed of differential mutation, crossover, and one-to-one selection. This design simplifies the genetic operations while preserving strong global search capabilities. Moreover, DE has an inherent adaptive mechanism that allows it to dynamically track the search progress and adjust its strategy accordingly, leading to high robustness and convergence efficiency.

At the core of the standard DE algorithm is a strategy where a new candidate solution is generated by adding the weighted difference between two randomly selected individuals to

a third individual. This mutant vector then undergoes crossover with a target vector, and the resulting trial vector is compared against the target. If the trial solution shows better fitness, it replaces the target in the next generation; otherwise, the target is retained. This iterative, fitness-driven selection mechanism ensures the survival of superior individuals and the gradual elimination of inferior ones, thereby guiding the population toward optimal solutions over time.

Compared to traditional optimization techniques [54,55], DE offers several distinct advantages:

1. Population-based search: It begins from multiple points (a population) rather than a single point, increasing the likelihood of locating the global optimum.

2. Derivative-free optimization: It relies solely on fitness values and does not require gradient information or continuity of the objective function, making it suitable for black-box and non-differentiable problems.

3. High parallelizability: Its intrinsic parallel structure makes it well-suited for large-scale distributed processing, reducing computational cost.

4. Probabilistic transition rules: DE operates based on stochastic search mechanisms rather than fixed deterministic rules, enhancing diversity and flexibility in the search space.

### 3.3 The basic flow of DE algorithm

The DE algorithm is a real-coded evolutionary algorithm. Its overall structure is similar to other evolutionary algorithms and comprises three fundamental operations: mutation [56], crossover [57], and selection [58]. The standard DE algorithm mainly includes the following four steps:

**1. Initialization of the population**

In an $n$-dimensional search space, generate $M$ individuals that satisfy the problem's constraints. The $i$-th individual is initialized as:

$$x_{ij}(0) = \text{rand}_{ij}(0,1) \cdot \left(x_{ij}^{\text{U}} - x_{ij}^{\text{L}}\right) + x_{ij}^{\text{L}} \tag{3}$$

Here, $x_{ij}^{\text{U}}$ and $x_{ij}^{\text{L}}$ denote the upper and lower bounds of the $j$-th component of the chromosome, respectively. $\text{rand}_{ij}(0,1)$ is a uniformly distributed random number in the range [0,1].

**2. Mutation operation**

For each individual, randomly select three distinct individuals from the current population, denoted as $x_{p1}$, $x_{p2}$, and $x_{p3}$, ensuring that $i \neq p_1 \neq p_2 \neq p_3$. The basic mutation operation is defined as:

$$h_{ij}(t+1) = x_{p1,j}(t) + F \cdot \left(x_{p2,j}(t) - x_{p3,j}(t)\right) \tag{4}$$

If no local optimization strategy is applied, an alternative mutation strategy that incorporates the current best individual can be expressed as:

$$h_{ij}(t+1) = x_{b,j}(t) + F \cdot \left(x_{p2,j}(t) - x_{p3,j}(t)\right) \tag{5}$$

Here, $(x_{p2,j}(t) - x_{p3,j}(t))$ is the difference vector, which is the core mechanism driving the exploration capability of DE. $F$ is the scaling factor, typically a constant in the range [0,2]. $x_{b,j}(t)$ represents the best individual in the current generation. This variant leverages the knowledge of the best solution so far to accelerate convergence.

**3. Crossover operation**

To increase population diversity, the DE algorithm performs a crossover operation between the original target vector and its corresponding mutant vector. The crossover is defined as:

$$v_{ij}(t+1) = \begin{cases} h_{ij}(t+1), & \text{if } \text{rand}_{ij} \leq \text{CR} \\ x_{ij}(t), & \text{otherwise} \end{cases} \tag{6}$$

Here, $\text{rand}_{ij}$ is a random number uniformly distributed in [0,1], and CR is the crossover probability, a user-defined parameter in the range [0,1]. This step ensures a combination of global and local search capabilities.

**4. Selection operation**

To decide whether the trial vector $v_i(t+1)$ should replace the current individual $x_i(t)$ in the next generation, the fitness function $f_a(\cdot)$ is used to compare the two:

$$x_i(t+1) = \begin{cases} v_i(t+1), & \text{if } f_a\left(v_{i1}(t+1), \ldots, v_{in}(t+1)\right) > f_a\left(x_{i1}(t), \ldots, x_{in}(t)\right) \\ x_i(t), & \text{otherwise} \end{cases} \tag{7}$$

Here, $f_a(\cdot)$ denotes the fitness function used to evaluate solution quality. Only those individuals that demonstrate better performance are retained in the population.

By repeatedly executing Step 2 through Step 4, the population evolves until a predefined maximum number of generations $G$ is reached. The overall operational flow of the differential evolution algorithm is illustrated in Fig 2.

## 3.4 The function optimization based on DE algorithm

In this section, we present a practical example of applying the DE algorithm to solve a function optimization problem, specifically targeting the Rosenbrock function to find its maximum, as shown in Fig 3. The function is defined as follows:

$$\begin{cases} f(x_1, x_2) = 100\left(x_1^2 - x_2\right)^2 + (1 - x_1)^2 \\ -2.048 \leq x_i \leq 2.048 \quad (i = 1, 2) \end{cases} \tag{8}$$

Within the specified domain, this function has two local maxima: $f(2.048, -2.048) =$ 3897.7342 and $f(-2.048, -2.048) = 3905.9262$, with the latter being the global maximum. As shown in the 3D plot of the function surface (see Fig 3), the two optima are relatively close in value and location, and the function exhibits significant nonlinearity and a narrow, curved valley structure.

To effectively locate the global maximum, we adopt the DE algorithm as a global optimization strategy. The fitness function is defined as $f_a(x) = f(x_1, x_2)$. Due to the presence of multiple local optima, it is crucial to guide the evolutionary process to avoid premature convergence.

This experiment, conducted as part of our study, illustrates the effectiveness of the DE algorithm in solving complex, multimodal optimization problems. The algorithm exhibited robust global search ability and convergence stability when applied to the Rosenbrock function, a well-known non-convex benchmark.

In this example, the maximum of the objective function is sought using real-valued encoding. Two real numbers are employed to represent the decision variables $x_1$ and $x_2$, respectively. The domain for each variable, [−2.048,2.048], is discretized into a set of Size equally spaced real values.

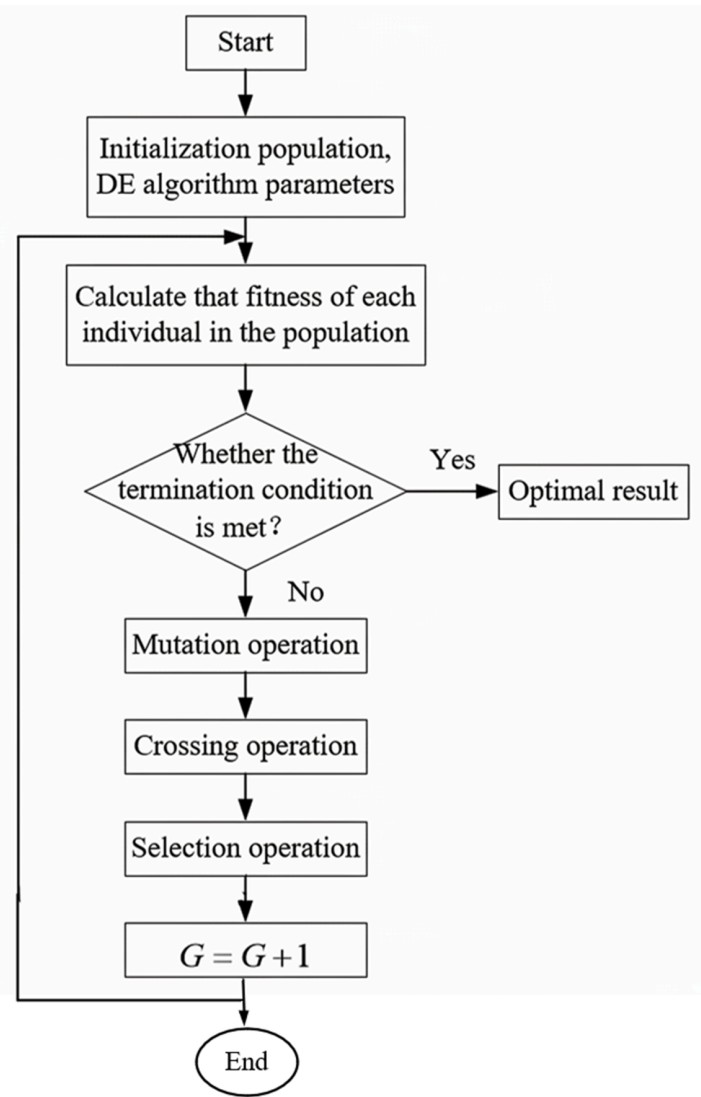

**Fig 2. The basic operation flow chart of DE.**

The fitness value of each individual is directly defined by the objective function itself, i.e., $f_a(x) = f(x_1, x_2)$, where the goal is to maximize the fitness.

The selection of these parameters is based on empirical studies and prior research on DE algorithm performance for complex nonlinear optimization problems. Specifically, the chosen mutation factor ($F = 1.2$) and crossover probability ($CR = 0.90$) fall within commonly recommended ranges for maintaining exploration capability and convergence stability [59]. The population size (50 individuals) and number of generations (30) are selected to ensure sufficient diversity and convergence within reasonable computational effort, consistent with previous applications of DE in multimodal function optimization tasks [60]. These parameter settings have also been preliminarily tested in simulation to avoid premature convergence. During the simulation of the Differential Evolution algorithm, the parameters are configured as follows (see Fig 4):

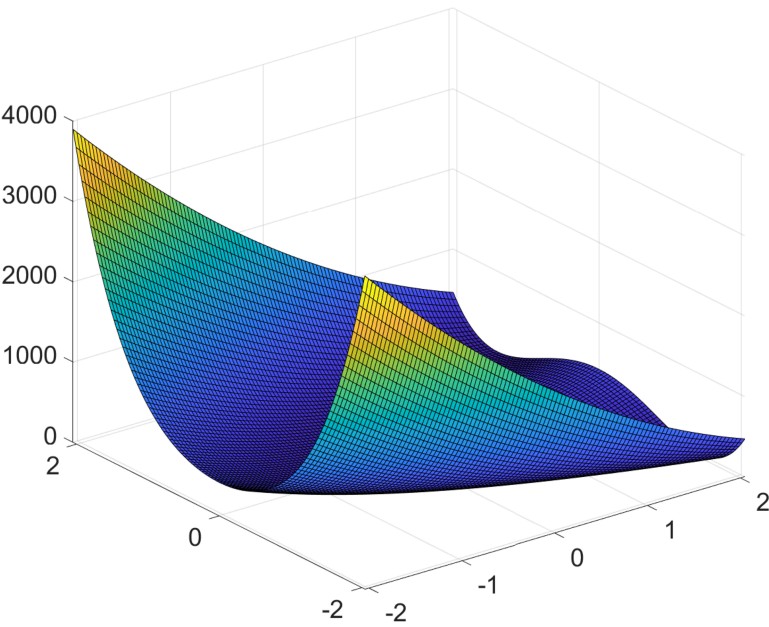

**Fig 3. Three-dimensional diagram of function $f(x_1, x_2)$.** The global maximum at (−2.048,−2.048) and a local maximum at (2.048,−2.048) are marked with red dots. These points highlight the multimodal nature of the objective function and illustrate the challenge of global optimization in a narrow curved valley structure.

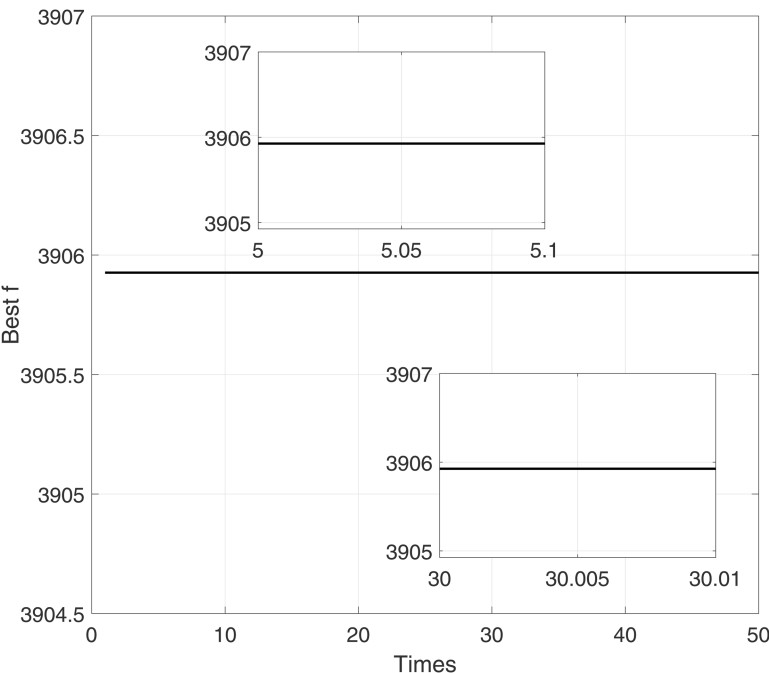

**Fig 4. Optimization process of fitness function $f_a(x)$.**

Mutation factor: $F$ = 1.2

Crossover probability: CR = 0.90

Population size: Size = 50

Maximum generations: $G$ = 30

Following the standard DE procedure (as described in Eqs 3 through 7), the algorithm evolves over 30 iterations and successfully identifies the global optimum solution:

$$\text{Best } S = [-2.048, -2.048]$$

At this point, the Rosenbrock function reaches its maximum value:

$$f(x_1, x_2) = 3905.92622$$

The convergence behavior of the fitness function is illustrated in Fig 4, showing that the DE algorithm exhibits strong global search capability and stable convergence. Moreover, the results confirm that increasing the mutation factor $F$ and expanding the population size are effective strategies to prevent premature convergence to local optima.

The simulation demonstrates a near 100% success rate in locating the global maximum, thereby validating the robustness and effectiveness of the implemented DE algorithm in solving this challenging optimization problem.

## 4 Simulation verification

### 4.1 Parameters setting of DE algorithm

To achieve optimal performance, the DE algorithm relies heavily on the appropriate setting of its control parameters. The effectiveness, convergence speed, and stability of the algorithm are all closely tied to these parameters. Moreover, for different types of optimization problems, the optimal parameter configurations may vary significantly. To enhance the convergence rate and robustness of DE, researchers have proposed numerous improvements and adaptive strategies, particularly focusing on the construction methods of the mutation vector—the core component of the algorithm.

The primary control parameters of the DE algorithm include [59] and [60]:

- Mutation factor ($F$);

- Crossover probability (CR);

- Population size ($M$);

- Maximum number of generations ($G$).

Let us elaborate on each parameter:

**1. Mutation Factor ($F$)**

The mutation factor $F$ plays a pivotal role in balancing population diversity and convergence behavior. It determines the amplitude of the differential variation vector, which directly affects how far a mutant vector can explore the solution space.

Typical Range: $F \in [0, 2]$

Recommended Range: $F = 0.3 \sim 0.6$

If $F$ is too small, the difference vector shrinks, leading to minimal perturbation and a high risk of premature convergence to local optima. If $F$ is too large, it enables global exploration but can significantly slow down convergence and destabilize the search process.

To address this trade-off, a linear scheduling strategy for $F$ is often adopted:

$$F = (F_{\max} - F_{\min}) \cdot \frac{T - t}{T} + F_{\min} \tag{9}$$

- $t$: current generation;
- $T$: maximum generation;
- $F_{max}$ and $F_{min}$: upper and lower bounds of the mutation factor.

This adaptive adjustment scheme allows for larger mutation steps in the early stages (promoting exploration), and smaller mutation steps in later stages (enhancing fine-tuning and convergence precision).

**2. Crossover Probability (CR)**

The crossover probability CR determines the likelihood of replacing elements from the parent vector with those from the mutant vector during the crossover phase. It governs the dimension-wise diversity and modulates the balance between local and global search.

- Typical Range: $CR \in [0, 1]$;
- Recommended Range: $CR = 0.6 \sim 0.9$.

A low **CR** implies fewer dimensions undergo crossover, which may reduce diversity and lead to early stagnation. A high CR increases the chance of exploring new regions but may cause loss of useful genetic material and slower convergence due to large perturbations exceeding the population's diversity range.

To enhance convergence dynamics, a linear adjustment strategy can also be applied to CR:

$$CR = CR_{\min} + \frac{(CR_{\max} - CR_{\min}) \cdot t}{T} \tag{10}$$

CRmin and CRmax are the minimum and maximum crossover probabilities.

By scheduling CR to increase over time, and $F$ to decrease, the DE algorithm is guided to focus on exploration at early stages and exploitation at later stages, which improves convergence without sacrificing diversity.

**3. Population Size ($M$)**

The population size $M$ refers to the number of candidate solutions maintained in each generation. It affects both the solution quality and the computational complexity of the algorithm.

Empirical Rule: $M \in [5D, 10D]$, where $D$ is the problem's dimensionality;

Minimum Value: $M \geq 4$ (to ensure valid mutation operations);

Typical Range: $M = 20 \sim 50$.

A larger population generally increases the diversity and the probability of discovering global optima, but also raises computation cost. A smaller population speeds up the process but risks premature convergence.

The optimal $M$ value should be selected based on:

Dimensionality of the problem;

Available computational resources;

Desired convergence speed and accuracy.

**4. Maximum number of generations ($G$)**

The maximum generation number $G$ is a stopping criterion that defines how long the evolutionary process should run.

A larger $G$ allows the algorithm more time to refine solutions, potentially achieving better accuracy;

However, longer iterations lead to increased computation time and may result in diminishing returns.

The choice of *G* should reflect:

Problem complexity;

Convergence behavior observed in preliminary runs;

Time constraints for optimization tasks.

The four core parameters—*F*, CR, *M*, and *G*—jointly determine the behavior, robustness, and performance of the Differential Evolution algorithm. Improper settings can lead to stagnation, slow convergence, or suboptimal results, whereas carefully tuned or adaptively adjusted parameters can significantly enhance the optimization performance.

In recent research, many adaptive or self-tuning DE variants have emerged, incorporating dynamic parameter control mechanisms that automatically adjust *F* and CR based on feedback from the evolutionary process, making DE more flexible and problem-independent.

As modern manufacturing rapidly advances toward digitalization and intelligent automation, scheduling optimization within production processes has become increasingly critical. In production environments where multiple machine tools operate in parallel, numerous batches of workpieces await processing, and delivery deadlines are stringent, determining how to efficiently arrange processing sequences to maximize resource utilization and economic benefits has become a core challenge for smart manufacturing.

Take a machining workshop as an example: it plans to process nine batches of different component orders using machine tools. Each component batch has a clearly defined delivery deadline. If processing is completed on time, the products can be sold at the contracted price, yielding standard profit. However, if the processing is delayed beyond the deadline, the products must be sold at a reduced price or may incur penalty costs, thereby impacting the total profit.

The objective of the scheduling plan is to identify the optimal processing sequence that maximizes the overall profit across all component orders. Each batch is modeled as a non-preemptive job (i.e., once a job starts, it must be completed without interruption). The processing time, delivery deadline, on-time profit, and overdue profit for each workpiece batch are shown in Table 1.

From Table 1, we can observe that the processing times for the product batches vary significantly, ranging from 1 to 7 units of time. In terms of delivery deadlines, product batches 1, 3, and 7 have relatively short deadlines (5, 3, and 5 units of time, respectively), while batches 6 and 8 have much later deadlines (24 and 6 units of time, respectively). This indicates that the scheduling must be optimized to ensure that products with tighter deadlines are prioritized without sacrificing the overall profit potential.

In terms of profit, product batches 8 and 9 stand out as they have the highest scheduled product profits of 5600 and 4500, respectively. However, they also have significant overdue product profits (4000 and 2000), which suggests that these batches are at risk of being delayed. The trade-off between on-time and overdue profits plays a crucial role in the optimization

**Table 1. Processing time, delivery deadline, and profit for each product batch.**

| Product batch | 1 | 2 | 3 | 4 | 5 | 6 | 7 | 8 | 9 |
|---|---|---|---|---|---|---|---|---|---|
| Processing time | 3 | 4 | 1 | 2 | 6 | 1 | 4 | 7 | 5 |
| Delivery deadline | 5 | 9 | 3 | 12 | 10 | 24 | 5 | 6 | 6 |
| Scheduled product profit | 750 | 1200 | 800 | 900 | 2500 | 500 | 3000 | 5600 | 4500 |
| Overdue product profit | 500 | 900 | 400 | 750 | 1800 | 300 | 1500 | 4000 | 2000 |

process, as the goal is to maximize the overall profit, considering both the scheduled and overdue profits.

This scheduling problem can be framed as a sequencing combinatorial optimization problem, which satisfies the delivery time constraint of each batch of parts and solves a set of optimal processing sequences to maximize the overall profit of parts.

## 4.2 Simulation parameter setting

In this machining scheduling simulation, the Differential Evolution (DE) algorithm employs real-valued encoding to represent processing sequences. The mutation factor is set to $F = 1.2$, the crossover probability to CR = 0.9, with a population size of 10 and a maximum of 100 generations. Each individual corresponds to a specific processing order, which is derived by sorting the real-valued vector to form a valid job sequence. The fitness function is defined as the total profit, calculated by comparing the cumulative processing time of each job against its delivery deadline. To ensure the validity of each sequence without duplication, functions such as unique, setdiff, and isempty are used for repair and verification. The entire simulation aims to maximize total profit and evaluate the effectiveness of the DE algorithm in solving complex scheduling optimization problems.

## 4.3 Simulation results show

Fig 5 illustrates the evolution of the best fitness value (i.e., total profit) in the population over 100 generations during the DE algorithm execution. The x-axis represents the number of generations, while the $y$-axis indicates the best profit value, measured in units of $10^4 RMB$. The initial population, consisting of 10 randomly generated job sequences, yields total profit

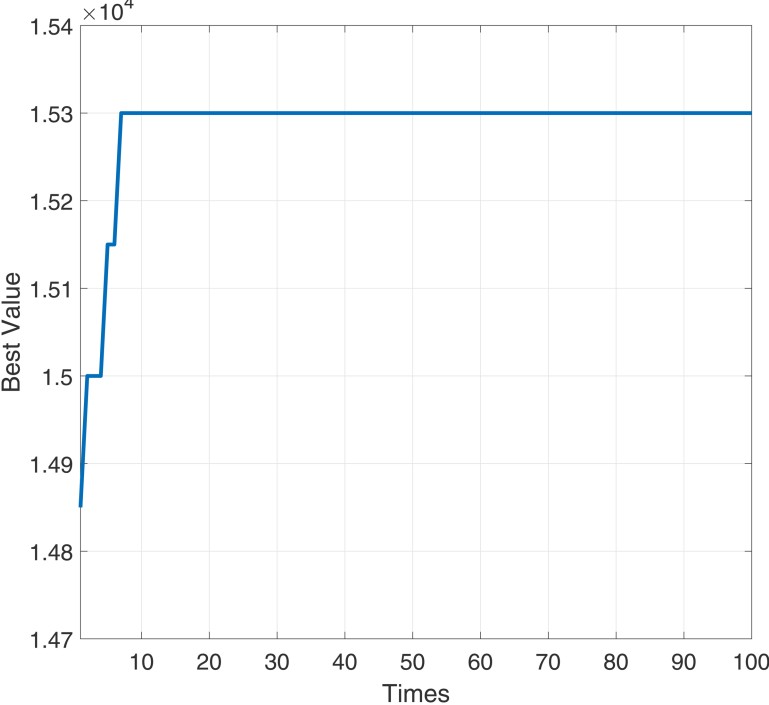

**Fig 5. Optimization process of total profit.**

values ranging from 14,700 to 15,000 RMB. During the first few generations, the algorithm quickly identifies better job sequences, and the total profit increases sharply from approximately 14,800 to over 15,200 RMB, indicating strong global exploration ability in the early phase. As the algorithm continues to evolve, the best profit value rapidly climbs to approximately 15,300 RMB, reaching a plateau after Generation 8. From Generation 10 onward, the best profit value remains stable around 15,300 RMB with no significant improvement. The algorithm has converged, and the solution found is likely to be globally optimal or very close to it. Fig 5 clearly demonstrates the effectiveness and convergence characteristics of the DE algorithm in solving the machining scheduling problem:

  - Fast Convergence: The optimal solution is found within the first 10 generations;

  - Stable Performance: The best value remains unchanged afterward, indicating algorithmic stability;

  - Strong Global Optimization Capability: A maximum profit of $15,300$ RMB is achieved, consistent with other independent simulation results, confirming that the algorithm did not fall into local optima;

  - Well-Tuned Parameters: The combination of a high mutation factor and crossover probability balances exploration and exploitation effectively.

## 5 Discussion and future work

This study demonstrates the effectiveness of the Differential Evolution (DE) algorithm in solving a single-machine scheduling problem with delivery deadlines and profit optimization constraints. By encoding job sequences as real-valued vectors and designing a fitness function based on on-time and delayed profits, the DE algorithm successfully identified optimal or near-optimal job sequences within a limited number of generations.

Simulation results show that the algorithm achieves rapid convergence (within the first 10 generations) and maintains high stability throughout the remaining iterations. The best total profit reached 15,300 RMB, confirming the algorithm's strong global search ability and its robustness in handling discrete combinatorial optimization problems through real-number encoding and appropriate repair mechanisms.

Furthermore, while the current study focuses on a single-machine scenario, the proposed DE-based approach has promising potential for extension to more complex manufacturing environments. In real-world production systems, challenges such as machine heterogeneity, sequence-dependent setup times, job priorities, and stochastic disruptions frequently arise. The real-coded representation and repair strategies used in this work can be adapted to accommodate such constraints. For instance, by modifying the encoding scheme and fitness function, the DE algorithm could be applied to flow shop or job shop scheduling problems. Additionally, hybridizing DE with local search or decomposition-based strategies would help scale the method to large-scale scheduling problems with high-dimensional decision spaces. These directions represent important steps toward making the proposed algorithm more practical and effective in real industrial applications.

## Supporting information

**S1 Text. Paper program.**
(PDF)

## Author contributions

**Conceptualization:** Yuehong Zhang, Mianhao Zhang.

**Data curation:** Yuehong Zhang.

**Formal analysis:** Yuehong Zhang.

**Project administration:** Mianhao Zhang.

**Resources:** Mianhao Zhang.

**Software:** Mianhao Zhang.

**Supervision:** Mianhao Zhang.

**Writing – original draft:** Yuehong Zhang.

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
