## [Decision Letter · Decision Letter 0]

5 Jun 2025

PONE-D-25-27253Optimization of machine tool processing scheduling based on differential evolution algorithmPLOS ONE

Dear Dr. Zhang,

Thank you for submitting your manuscript to PLOS ONE. After careful consideration, we feel that it has merit but does not fully meet PLOS ONE’s publication criteria as it currently stands. Therefore, we invite you to submit a revised version of the manuscript that addresses the points raised during the review process.

We look forward to receiving your revised manuscript.

Kind regards,

Syed Hamid Hussain Madni

Academic Editor

PLOS ONE

Journal Requirements:

2. Please ensure that you refer to Figure 1, 2 and 4 in your text as, if accepted, production will need this reference to link the reader to the figure.

Reviewers' comments:

Reviewer's Responses to Questions

**Comments to the Author**

1. Is the manuscript technically sound, and do the data support the conclusions?

Reviewer #1: Yes

Reviewer #2: Yes

2. Has the statistical analysis been performed appropriately and rigorously? 

Reviewer #1: No

Reviewer #2: Yes

3. Have the authors made all data underlying the findings in their manuscript fully available?

Reviewer #1: No

Reviewer #2: Yes

4. Is the manuscript presented in an intelligible fashion and written in standard English?

Reviewer #1: Yes

Reviewer #2: Yes

5. Review Comments to the Author

Reviewer #1: 1) It would be beneficial to separate the introduction section from the related work. Additionally, critically analyze individual related papers that are closely aligned with your proposed method. Highlight their strengths and weaknesses, particularly in addressing the specific problem, to clearly demonstrate how the research gap was identified.

2) Revise the methodology section by including a block diagram or flowchart that clearly illustrates the DE algorithm and highlights the hybrid components constituting your main contribution. Keep the description of the standard DE algorithm concise, focusing more on your proposed modifications.

3) Subsection 3.4, which discusses the parameter settings of the DE algorithm, should be moved under the "Experiments and Results" section. Clearly describe the experimental platform and the corresponding parameter configurations.

4) Present the experimental results in tabular form, including the evaluation metrics used. Supplement the tables with appropriate statistical visualizations and provide a thorough discussion of the results.

5) Clearly elaborate on the adaptation of the DE algorithm to discrete scheduling tasks through custom encoding and decoding schemes, emphasizing this as a key contribution of your work.

6) It would strengthen the practical relevance of your proposed method if it were validated using a real-world industrial dataset. Additionally, focus on conducting a deeper analysis of the experimental results.

7) The multiple objectives mentioned in the manuscript appear to be unnecessary, as the main focus of this work is on profit maximization as a single objective.

8) Simplify the assumptions made in relation to the main objective. The model currently depends on simplifications such as fixed switching times, idealized precedence constraints, and homogeneous job sizes, which may limit its applicability in complex manufacturing environments.

9) To better demonstrate the superiority of your proposed method, consider benchmarking your DE-based approach against modern metaheuristics such as NSGA-II, MOEA/D, and ACD.

10) Further discussion on the empirical performance of the proposed method in handling complex scheduling instances, particularly in the context of manufacturing, would significantly strengthen your paper.

Reviewer #2: 1. HEADING 2 have no content, need to add at least a paragraph under this heading

2. Heading 2.2 , the adjustment of variables’ was done on what basis, either add explanation or include citation

3. Heading 3 also have no content, just like heading 2

4. In figure 2 the “feedback loop” must come after the last block “G=G+1

5. FOR THE HEADING “3.4 Parameters setting of DE algorithm” reference/citation is missing for their value selection

6. Figure 3 must be more explanatory, should show the local maxima and minima in the fig

7. Lines 395 and 398 , what examples? Are they examples or your contribution? If it’s your then rephrase the sentence.

8. Again at line 404 and next the values of parameters have been selected on what criteria? Kindly specify

9. Figure 4 must be cited in the text and add some explanations about it.

10. Table 1 better to add some discussion on table findings

11. Heading 4.1 again the reference or explanation is required to added for parameters setting

12. Better to add those points, discuss about the achieved tasks, mentioned in gaps section

13. Must add the recent references, the latest reference, added is from year 2021

6. PLOS authors have the option to publish the peer review history of their article (what does this mean?). If published, this will include your full peer review and any attached files.

Reviewer #1: **Yes: **Ayuba John

Reviewer #2: No

---

## [Author Response · Author response to Decision Letter 1]

16 Jun 2025

Peer-to-peer reviewers can see the attachment uploaded by the system.

---

## [Decision Letter · Decision Letter 1]

18 Sep 2025

Optimization of machine tool processing scheduling based on differential evolution algorithm

PONE-D-25-27253R1

Dear Dr. Zhang,

We’re pleased to inform you that your manuscript has been judged scientifically suitable for publication and will be formally accepted for publication once it meets all outstanding technical requirements.

Kind regards,

Syed Hamid Hussain Madni

Academic Editor

PLOS ONE

Additional Editor Comments (optional):

Reviewer #1:

Reviewer #2:

Reviewers' comments:

Reviewer's Responses to Questions

**Comments to the Author**

1. If the authors have adequately addressed your comments raised in a previous round of review and you feel that this manuscript is now acceptable for publication, you may indicate that here to bypass the “Comments to the Author” section, enter your conflict of interest statement in the “Confidential to Editor” section, and submit your "Accept" recommendation.

Reviewer #1: All comments have been addressed

Reviewer #2: All comments have been addressed

2. Is the manuscript technically sound, and do the data support the conclusions?

Reviewer #1: Yes

Reviewer #2: (No Response)

3. Has the statistical analysis been performed appropriately and rigorously? 

Reviewer #1: Yes

Reviewer #2: (No Response)

4. Have the authors made all data underlying the findings in their manuscript fully available?

Reviewer #1: Yes

Reviewer #2: Yes

5. Is the manuscript presented in an intelligible fashion and written in standard English?

Reviewer #1: Yes

Reviewer #2: Yes

6. Review Comments to the Author

Reviewer #1: 1) It is quite impressive that the authors have substantially addressed the issues raised in the reviewers’ comments on the original manuscript. Their revisions have significantly improved the quality of the manuscript, bringing it up to the standard required by this reputable journal. I therefore confidently recommend that the manuscript be accepted for publication.

2) You might find these papers useful, and citing them could strengthen your manuscript:

http://doi.org/10.1049/wss2.12100

https://doi.org/10.1016/j.iswa.2024.200381

https://doi.org/10.1016/j.iswa.2024.200442

Reviewer #2: author has adressed the previously mentioned comments.

previously i have mentioned some of these comments (1. HEADING 2 have no content, need to add at least a paragraph under this heading

2. Heading 2.2 , the adjustment of variables’ was done on what basis, either add explanation or include citation

3. Heading 3 also have no content, just like heading 2

4. In figure 2 the “feedback loop” must come after the last block “G=G+1

5. FOR THE HEADING “3.4 Parameters setting of DE algorithm” reference/citation is missing for their value selection

6. Figure 3 must be more explanatory, should show the local maxima and minima in the fig

7. Lines 395 and 398 , what examples? Are they examples or your contribution? If it’s your then rephrase the sentence.

8. Again at line 404 and next the values of parameters have been selected on what criteria? Kindly specify

9. Figure 4 must be cited in the text and add some explanations about it.

10. Table 1 better to add some discussion on table findings

11. Heading 4.1 again the reference or explanation is required to added for parameters setting

12. Better to add those points, discuss about the achieved tasks, mentioned in gaps section

13. Must add the recent references, the latest reference, added is from year 2021

).

so there is no need of any revision

7. PLOS authors have the option to publish the peer review history of their article (what does this mean?). If published, this will include your full peer review and any attached files.

Reviewer #1: **Yes: **Ayuba John

Reviewer #2: No

---

## [Editor Report · Acceptance letter]

PONE-D-25-27253R1

PLOS ONE

Dear Dr. Zhang,

I'm pleased to inform you that your manuscript has been deemed suitable for publication in PLOS ONE. Congratulations! Your manuscript is now being handed over to our production team.

Kind regards,

on behalf of

Dr. Syed Hamid Hussain Madni

Academic Editor

PLOS ONE